# Lower Bounds for Differentially Private ERM: Unconstrained and Non-Euclidean

## Abstract

We consider the lower bounds of differentially private empirical risk minimization (DP-ERM) for convex functions in both constrained and unconstrained cases concerning the general $\ell_p$ norm beyond the $\ell_2$ norm considered by most of the previous works.

We provide a simple black-box reduction approach that can generalize lower bounds in constrained to unconstrained cases. Moreover, for $(\epsilon, \delta)$-DP, we achieve the optimal $\Omega(\frac{\sqrt{d \log(1/\delta)}}{\epsilon n})$ lower bounds for both constrained and unconstrained cases and any $\ell_p$ geometry where $p \geq 1$ by considering $\ell_1$ loss over the $\ell_\infty$ ball.

## 1 Introduction

Since the seminal work of Dwork et al. (2006), differential privacy (DP), defined below, has become the standard and rigorous notion of privacy guarantee for machine learning algorithms.

**Definition 1.1** (Differential privacy). A randomized mechanism $\mathcal{M}$ is $(\epsilon, \delta)$-differentially private[1] if for any event $\mathcal{O} \in \text{Range}(\mathcal{M})$ and for any neighboring databases $\mathcal{D}$ and $\mathcal{D}'$ that differ by a single data element, one has

$$\Pr[\mathcal{M}(\mathcal{D}) \in \mathcal{O}] \leq \exp(\epsilon) \Pr[\mathcal{M}(\mathcal{D}') \in \mathcal{O}] + \delta.$$

Among the rich literature on DP, many fundamental problems are based on empirical risk minimization (ERM), and DP-ERM becomes one of the most well-studied problems in the DP community. See e.g., Chaudhuri & Monteleoni (2008); Rubinstein et al. (2009); Chaudhuri et al. (2011); Kifer et al. (2012); Song et al. (2013); Bassily et al. (2014); Jain & Thakurta (2014); Talwar et al. (2015); Kasiviswanathan & Jin (2016); Fukuchi et al. (2017); Wu et al. (2017); Zhang et al. (2017); Wang et al. (2017); Iyengar et al. (2019); Bassily et al. (2020); Kulkarni et al. (2021); Asi et al. (2021); Bassily et al. (2021b); Wang et al. (2021); Bassily et al. (2021a); Gopi et al. (2022); Arora et al. (2022); Ganesh et al. (2022).

In DP-ERM, we are given a family convex functions where each function $\ell(\cdot; z)$ is defined on a convex set $\mathcal{K} \subseteq \mathbb{R}^d$, and a data-set $\mathcal{D} = \{z_1, \cdots, z_n\}$ to design a differentially private algorithm that can minimize the loss function

$$L(\theta; \mathcal{D}) = \frac{1}{n} \sum_{i=1}^{n} \ell(\theta; z_i), \tag{1}$$

and the value $L(\theta; \mathcal{D}) - \min_{\theta' \in \mathcal{K}} L(\theta'; \mathcal{D})$ is called the excess empirical loss with respect to solution $\theta$, measuring how it compares with the best solution in $\mathcal{K}$.

DP-ERM in the constrained case and Euclidean geometry (with respect to $\ell_2$ norm) was studied first, well-studied, and most of the previous literature belongs to this case. More specifically, the Euclidean constrained case considers convex loss functions defined on a bounded convex set $\mathcal{C} \subsetneq \mathbb{R}^d$, assuming the functions are 1-Lipschitz over the convex set of diameter 1 with respect to the $\ell_2$ norm. For pure-DP (i.e. $(\epsilon, 0)$-DP), the seminal work Bassily et al. (2014) achieved tight upper and lower bounds $\Theta(\frac{d}{\epsilon n})$. As for approximate-DP (i.e. $(\epsilon, \delta)$-DP when $\delta > 0$), previous works Bassily et al.

---

[1] When $\delta > 0$, we may refer to it as approximate-DP, and we name the particular case when $\delta = 0$ pure-DP sometimes.

(2014); Steinke & Ullman (2016); Wang et al. (2017); Bassily et al. (2019) achieved the tight bound $\Theta(\frac{\sqrt{d\log(1/\delta)}}{\epsilon n})$.

DP-ERM in the unconstrained case was neglected before and gathered people's attention recently. Jain & Thakurta (2014); Song et al. (2021) found a tight bound $\tilde{O}(\frac{\sqrt{\text{rank}}}{\epsilon n})$ for minimizing the excess empirical risk of Generalized Linear Models (GLMs, see Definition A.1 in Appendix) in the unconstrained case and evaded the curse of dimensionality, where $\text{rank}$ is the rank of the feature matrix in the GLM problem. As a comparison, the tight bound $\tilde{\Theta}(\frac{\sqrt{d}}{\epsilon n})$ holds for the constrained DP-GLM, even for the overparameterized case when $\text{rank} \le n \ll d$. The dimension-independent result is intriguing, as modern machine learning models are usually huge, with millions to billions of parameters (dimensions).

A natural question arises whether one can get similar dimension-independent results for a more general family of functions beyond GLMs. Unfortunately, Asi et al. (2021) provided a negative answer and gave an $\Omega(\frac{\sqrt{d}}{n\epsilon\log d})$ lower bound for some general convex functions. Their method chooses appropriate objective functions and utilizes one-way marginals, but the extra logarithmic term in their bound seems nontrivial to remove in the unconstrained case.

Another aspect is DP-ERM in non-Euclidean settings. Most previous works in the literature consider the constrained Euclidean setting where the convex domain and (sub)gradients of objective functions have bounded $\ell_2$ norms, and DP-ERM concerning the general $\ell_p$ norm is much less well-understood. Motivated by the importance and wide applications of non-Euclidean settings, some previous works Talwar et al. (2015); Asi et al. (2021); Bassily et al. (2021b) analyzed constrained DP-ERM with respect to the general $\ell_p$ norm with many exciting results, and there is still room for improvement in many regimes.

| Article | Constrained? | $\ell_p$ | Loss Function | Pure DP | Approximate DP |
|---------|-------------|----------|---------------|---------|----------------|
| Bassily et al. (2014) | constrained | $p = 2$ | GLM | $\Omega(\frac{d}{n\epsilon})$ | $\Omega(\frac{\sqrt{d}}{n\epsilon})$ |
| Steinke & Ullman (2016) | constrained | $\ell_2$ | GLM | N/A | $\Omega(\frac{\sqrt{d\log(1/\delta)}}{n\epsilon})$ |
| Song et al. (2021) | unconstrained | $p = 2$ | GLM | N/A | $\Omega(\frac{\sqrt{\text{rank}}}{n\epsilon})$ |
| Asi et al. (2021) | both | $p = 1$ | general | N/A | $\Omega(\frac{\sqrt{d}}{n\epsilon\log d})$ |
| Bassily et al. (2021b) | constrained | $1 < p \le 2$ | GLM | N/A | $\Omega((p-1)\frac{\sqrt{d\log(1/\delta)}}{n\epsilon})$ |
| Ours | both | $1 \le p \le \infty$ | general | $\Omega(\frac{d}{n\epsilon})$ | $\Omega(\frac{\sqrt{d\log(1/\delta)}}{n\epsilon})$ |

Table 1: Comparison of lower bounds for private convex ERM. One can easily extend our lower bounds in the unconstrained case to the constrained case. The lower bound of Song et al. (2021) is weaker than ours in the important over-parameterized $d \gg n$ setting, as $\text{rank} \le \min\{n, d\}$.

## 1.1 OUR CONTRIBUTIONS

This paper considers the lower bounds for DP-ERM under unconstrained and/or non-euclidean settings. We summarize our main results as follows:

- We propose a black-box reduction approach, which directly generalizes the lower bounds in constrained cases to the unconstrained case. Such a method is beneficial for its simplicity. Nearly all exiting lower bounds in the constrained case can be extended to the unconstrained case directly, and any new progress in the constrained case can be of immediate use due to its black-box nature.

- We achieve $\Omega(\frac{\sqrt{d\log(1/\delta)}}{\epsilon n})$ lower bounds for both constrained and unconstrained cases and any $\ell_p$ geometry for $p \ge 1$ at the same time by considering $\ell_1$ loss over the $\ell_\infty$ ball. This bound improves previous results, exactly matches the upper bounds for $1 < p \le 2$ and obtains novel bounds for $p > 2$.[2]

---

[2]The current best upper bound is $O(\min\{\log d, \frac{1}{p-1}\}\frac{\sqrt{d\log(1/\delta)}}{\epsilon n})$ for $1 \le p \le 2$ and $O(\frac{d^{1-1/p}\sqrt{\log(1/\delta)}}{\epsilon n})$ for $2 < p \le \infty$.

As an example of the application of our simple reduction approach, we will show how to get the $\Omega(\frac{d}{\epsilon n})$ lower bound in the unconstrained pure-DP case from the result in constrained case Bassily et al. (2014), which is the first lower bound in this setting to the best of our knowledge. This reduction also demonstrates that evading the curse of dimensionality is impossible based on existing dimension-dependent lower bounds, even in the (arguably less complicated) unconstrained case.

## 1.2 RELATED WORK

Previous studies on DP-ERM primarily focus on the constrained setting. The unconstrained case recently attracted people's interest because Jain & Thakurta (2014); Song et al. (2021) found an $O(\frac{\sqrt{\text{rank}}}{\epsilon n})$ upper bound for minimizing the excess risk of GLMs, which evades the curse of dimensionality. It has been known that the constrained condition plays a crucial role in achieving dimension independence, as pointed out by the $\Omega(\frac{\sqrt{d}}{n\epsilon})$ lower bound in Bassily et al. (2014), particularly for minimizing constrained GLMs for the case when "$\text{rank} \leq n \ll d$". There are fundamental differences between constrained and unconstrained cases, and analyzing the unconstrained case is an important direction.

Most existing lower bounds of DP-ERM use GLM functions. For example, the objective function used in Bassily et al. (2014) is a linear function $\ell(\theta; z) = \langle \theta, z \rangle$ which cannot be applied in the unconstrained case; otherwise, the loss value would be infinite. Considering this limitation, Song et al. (2021) adopted the objective functions $\ell(\theta; z) = |\langle \theta, x \rangle - y|$. They transferred the problem of minimizing GLM to estimating the mean of a set of vectors, then got the lower bound by tools from the coding theory.

Kairouz et al. (2020); Zhou et al. (2020) considered how to evade the curse of dimensionality for more general functions beyond GLMs with public data, which serves to identify a low-rank subspace similar to Song et al. (2021) in spirit. DP Stochastic Convex Optimization (SCO) Feldman et al. (2020); Bassily et al. (2020; 2019); Kulkarni et al. (2021); Asi et al. (2021); Bassily et al. (2021b) is another important and fundamental problem which is closely related to DP-ERM. Roughly speaking, in the SCO, the objective is to minimize the function $\mathbb{E}_{z \sim \mathcal{P}}[\ell(\theta; z)]$ for some underlying distribution $\mathcal{P}$, which requires analyses on the generalization ability of the algorithms. The tight bound of DP-SCO is usually the maximum among the informational lower bound on (non-private) SCO and the lower bound on DP-ERM, and improved lower bounds on DP-ERM can also benefit research on DP-SCO.

## 2 PRELIMINARY

We introduce the prior background knowledge required in the rest of the paper. Additional background knowledge, such as the definition of GLM, can be found in the appendix. We start with defining Lipschitz functions.

**Definition 2.1** (G-Lipschitz Continuity). *A function $f : \mathcal{K} \to \mathbb{R}$ is G-Lipschitz continuous with respect to $\ell_p$ geometry if for all $\theta, \theta' \in \mathcal{K}$, one has:*

$$|f(\theta) - f(\theta')| \leq G\|\theta - \theta'\|_p. \tag{2}$$

### 2.1 PROPERTIES OF DIFFERENTIAL PRIVACY

In this subsection, we introduce several very basic properties of differential privacy without proving them (refer Dwork et al. (2014) for details).

**Proposition 2.2** (Group privacy). *If $\mathcal{M} : X^n \to Y$ is $(\epsilon, \delta)$-differentially private mechanism, then for all pairs of datasets $x, x' \in X^n$, then $\mathcal{M}(x), \mathcal{M}(x')$ are $(k\epsilon, k\delta e^{k\epsilon})$-indistinguishable when $x, x'$ differs on at most $k$ locations.*

**Proposition 2.3** (Post processing). *If $\mathcal{M} : X^n \to Y$ is $(\epsilon, \delta)$-differentially private and $\mathcal{A} : Y \to Z$ is any randomized function, then $\mathcal{A} \circ \mathcal{M} : X^n \to Z$ is also $(\epsilon, \delta)$-differentially private.*

## 3 REDUCTION APPROACH

In this section, we prove a general black-box reduction theorem, which can directly generalize the lower bounds on DP-ERM in the constrained case to the unconstrained case. As an application, we give an example for using our reduction approach to get pure-DP lower bound in the unconstrained case from the constrained case Bassily et al. (2014).

To begin with, we introduce the following lemma from Cobzas & Mustata (1978), which gives a Lipschitz extension of any convex Lipschitz function over some bounded convex set to the whole domain $\mathbb{R}^d$.

**Lemma 3.1** (Theorem 1 in Cobzas & Mustata (1978)). *Let $f$ be a convex function which is $\eta$-Lipschitz w.r.t. $\ell_2$ and defined on a convex bounded set $\mathcal{K} \subset \mathbb{R}^d$. Define an auxiliary function $g_y(x)$ as:*

$$g_y(x) := f(y) + \eta\|x - y\|_2, y \in \mathcal{K}, \forall x \in \mathbb{R}^d. \tag{3}$$

*Then consider the function $\tilde{f} : \mathbb{R}^d \to \mathbb{R}$ defined as $\tilde{f}(x) := \min_{y\in\mathcal{K}} g_y(x)$. We know $\tilde{f}$ is $\eta$-Lipschitz w.r.t. $\ell_2$ on $\mathbb{R}^d$, and $\tilde{f}(x) = f(x)$ for any $x \in \mathcal{K}$.*

For any $y \in \mathbb{R}^d$, we define $\Pi_{\mathcal{K}}(y) := \arg\min_{x\in\mathcal{K}} \|x - y\|_2$. It is well-known in the convex analysis, that for a compact convex set $\mathcal{K}$ and any point $y \in \mathbb{R}^d$, the the set $\{x \in \mathcal{K} : \|x-y\|_2 < \|z-y\|_2, \forall z \in \mathcal{K}, z \neq x\}$ is always non-empty and singleton Hazan (2019). In short, to prove lower bounds for the unconstrained case, one can extend the loss function in the constrained domain to $\mathbb{R}^d$ with an important observation on such convex extension: the loss $L(\theta; \mathcal{D})$ at a point $\theta$ does not increase after projecting $\theta$ to the convex domain $\mathcal{K}$, i.e. $L(\theta; \mathcal{D}) \geq L(\Pi_{\mathcal{K}}(\theta); \mathcal{D})$. One can derive this property from the Pythagorean Theorem (Lemma 3.2) for any convex set by combining with the particular structure of the extension.

**Lemma 3.2** (Pythagorean Theorem for convex set). *Letting $\mathcal{K} \subset \mathbb{R}^d$ be a convex set, $y \in \mathbb{R}^d$ and $x = \Pi_{\mathcal{K}}(y)$, then for any $z \in \mathcal{K}$ we have:*

$$\|x - z\|_2 \leq \|y - z\|_2. \tag{4}$$

In the unconstrained case, we usually assume a public prior knowledge on $C$ where $C \geq \|\theta_0 - \theta^*\|_2$, $\theta_0$ is the public initial point and $\theta^*$ is the optimal solution to $L(\cdot; \mathcal{D})$ over $\mathbb{R}^d$.

To proceed, we first assume some lower bound in the constrained case, which we use to reduce. The definition below defines a witness function for any lower bound in the constrained case, for example in Bassily et al. (2014) the (witness) loss function is simply linear and the lower bound is roughly $\Omega(\min\{1, \frac{\sqrt{d}}{n\epsilon}\})$.

**Definition 3.3.** Let $n, d$ be large enough, $0 \leq \delta \leq 1$ and $\epsilon > 0$. We say functions $\ell$ is a witness to the lower bound function $f$, if for any $(\epsilon, \delta)$-DP algorithm, there exist a convex set $\mathcal{K} \subset \mathbb{R}^d$ of diameter $C$, a family of $G$-Lipschitz convex functions $\ell(\theta; z)$ defined on $\mathcal{K}$ w.r.t. $\ell_2$, a dataset $\mathcal{D}$ of size $n$, such that with probability at least $1/2$ (over the random coins of the algorithm),

$$L(\theta^{priv}; \mathcal{D}) - \min_{\theta\in\mathcal{K}} L(\theta; \mathcal{D}) = \Omega(f(d, n, \epsilon, \delta, G, C)),$$

where $L(\theta; \mathcal{D}) := \frac{1}{n}\sum_{i=1}^n \ell(\theta; z_i)$ and $\theta^{priv} \in \mathcal{K}$ is the output of the algorithm.

The function $f$ can be any lower bound in the constrained case with dependence on the parameters, and $\ell$ is the loss function used to construct the lower bound. We use the Lipschitz extension mentioned above to define our loss function in the unconstrained case, i.e.,

$$\tilde{\ell}(\theta; z) = \min_{y\in\mathcal{K}} \ell(y; z) + G\|\theta - y\|_2 \tag{5}$$

which is convex, G-Lipschitz and equal to $\ell(\theta; z)$ when $\theta \in \mathcal{K}$ by Lemma 3.1. Our intuition is simple: if $\theta^{priv}$ lies in $\mathcal{K}$, then we are done by using the witness function and lower bound from Definition 3.3. If not, the projection of $\theta^{priv}$ to $\mathcal{K}$ should lead to a smaller loss. However, the projected point cannot have a minimal loss due to the lower bound in Definition 3.3, let alone $\theta^{priv}$ itself. We have the following theorem that achieves the general reduction approach in the unconstrained setting:

**Theorem 3.4.** *Assume $\ell$, $f$ are the witness function and lower bound as in Definition 3.3. For any $(\epsilon, \delta)$-DP algorithm and any initial point $\theta_0 \in \mathbb{R}^d$, there exist a family of $G$-Lipschitz convex functions $\tilde{\ell}(\theta; z) : \mathbb{R}^d \to \mathbb{R}$ being the $\ell$ from Definition 3.3, a dataset $\mathcal{D}$ of size $n$ and the same function $f$, such that with probability at least $1/2$ (over the random coins of the algorithm)*

$$\tilde{L}(\theta^{priv}; \mathcal{D}) - \tilde{L}(\theta^*; \mathcal{D}) = \Omega(f(d, n, \epsilon, \delta, G, C)), \tag{6}$$

*where $\tilde{L}(\theta; \mathcal{D}) := \frac{1}{n} \sum_{z_i \in \mathcal{D}} \tilde{\ell}(\theta; z_i)$ is the ERM objective function, $\theta^* = \arg\min_{\theta \in \mathbb{R}^d} \tilde{L}(\theta; \mathcal{D})$, $C \geq \|\theta_0 - \theta^*\|_2$ and $\theta^{priv}$ is the output of the algorithm.*

*Proof.* Without loss of generality, let $\mathcal{K} = \{\theta : \|\theta - \theta_0\|_2 \leq C\}$ be the $\ell_2$ ball around $\theta_0$, let $\ell(\theta; z)$ be the convex functions used in Definition 3.3, and as mentioned we can find our loss functions $\tilde{\ell}(\theta; z) = \min_{y \in \mathcal{K}} \ell(y; z) + G\|\theta - y\|_2$. As $\theta^* \in \mathcal{K}$, we know that

$$\tilde{L}(\theta^*; \mathcal{D}) = \min_{\theta \in \mathcal{K}} L(\theta; D). \tag{7}$$

Denote $\tilde{\theta}^{priv} = \Pi_{\mathcal{K}}(\theta^{priv})$ the projected point of $\theta^{priv}$ to $\mathcal{K}$. Because post-processing keeps privacy, outputting $\tilde{\theta}^{priv}$ is also $(\epsilon, \delta)$-DP. By Definition 3.3, we have

$$L(\tilde{\theta}^{priv}; \mathcal{D}) - \min_{\theta} L(\theta; \mathcal{D}) = \Omega(f(d, n, \epsilon, \delta, G, C)). \tag{8}$$

If $\tilde{\theta}^{priv} = \theta^{priv}$, which means $\theta^{priv} \in \mathcal{K}$, then because $\tilde{\ell}(\theta; z)$ is equal to $\ell(\theta; z)$ for any $\theta \in \mathcal{K}$ and $z$, one has $\tilde{L}(\theta^{priv}; \mathcal{D}) = \tilde{L}(\tilde{\theta}^{priv}; \mathcal{D}) = L(\tilde{\theta}^{priv}; \mathcal{D})$.

If $\tilde{\theta}^{priv} \neq \theta^{priv}$ which means $\theta^{priv} \notin \mathcal{K}$, then since $\ell(\cdot; z)$ is $G$-Lipschitz, for any $z$, we have that (denoting $y^* = \arg\min_{y \in \mathcal{K}} \ell(y; z) + G\|\theta^{priv} - y\|_2$):

$$\begin{aligned}
\tilde{\ell}(\theta^{priv}; z) &= \min_{y \in \mathcal{K}} \ell(y; z) + G\|\theta^{priv} - y\|_2 \\
&= \ell(y^*; z) + G\|\theta^{priv} - y^*\|_2 \\
&\geq \ell(y^*; z) + G\|\tilde{\theta}^{priv} - y^*\|_2 \\
&\geq \min_{y \in \mathcal{K}} \ell(y; z) + G\|\tilde{\theta}^{priv} - y\|_2 \\
&= \tilde{\ell}(\tilde{\theta}^{priv}; z),
\end{aligned}$$

where the third line is by the Pythagorean Theorem for the convex set, see Lemma 3.2. We have $\tilde{L}(\theta^{priv}; \mathcal{D}) \geq \tilde{L}(\tilde{\theta}^{priv}; \mathcal{D}) = L(\tilde{\theta}^{priv}; \mathcal{D})$. In a word, we get

$$\tilde{L}(\theta^{priv}; \mathcal{D}) \geq \tilde{L}(\tilde{\theta}^{priv}; \mathcal{D}) = L(\tilde{\theta}^{priv}; \mathcal{D}). \tag{9}$$

Combining Equation (7), (8) and (9) together, we have that

$$\begin{aligned}
&\tilde{L}(\theta^{priv}; \mathcal{D}) - \tilde{L}(\theta^*; \mathcal{D}) \\
&= \tilde{L}(\theta^{priv}; \mathcal{D}) - \min_{\theta} L(\theta; \mathcal{D}) \\
&\geq L(\tilde{\theta}^{priv}; \mathcal{D}) - \min_{\theta} L(\theta; \mathcal{D}) \\
&\geq \Omega(f(d, n, \epsilon, \delta, G, C)).
\end{aligned}$$

$\square$

### 3.1 EXAMPLE FOR PURE-DP

This subsection gives a concrete example of the reduction method in the pure-DP setting. In the construction of lower bounds for constrained DP-ERM in Bassily et al. (2014), they chose the linear function $\ell(\theta; z) = \langle \theta, z \rangle$ as the objective function, which is not applicable in the unconstrained setting because it could decrease to negative infinity. Instead, we extend the linear loss in unit $\ell_2$ ball to the whole $\mathbb{R}^d$ while preserving its Lipschitzness and convexity. We use such an extension to define our loss function in the unconstrained case. Namely, we define

$$\ell(\theta; z) = \min_{\|y\|_2 \leq 1} -\langle y, z \rangle + \|\theta - y\|_2 \tag{10}$$

for all $\theta, z$ in the unit $\ell_2$ ball, which is convex, 1-Lipschitz and equal to $-\langle \theta, z \rangle$ when $\|\theta\|_2 \leq 1$ according to Lemma 3.1. Specifically, it's easy to verify that $\ell(\theta; 0) = \max\{0, \|\theta\|_2 - 1\}$. When $\|z\|_2 = 1$, one has

$$\ell(\theta; z) \geq \min_{\|y\|_2 \leq 1} -\langle y, z \rangle \geq -1, \tag{11}$$

where the equation holds if and only if $\theta = z$.

For any dataset $\mathcal{D} = \{z_1, ..., z_n\}$, we define $L(\theta; \mathcal{D}) = \frac{1}{n} \sum_{i=1}^{n} \ell(\theta; z_i)$. We need the following lemma from Bassily et al. (2014) to prove the lower bound. The proof is similar to that of Lemma 5.1 in Bassily et al. (2014), except that we change the construction by adding points $\mathbf{0}$ (the all-zero $d$ dimensional vector) as our dummy points. For completeness, we include it here.

**Lemma 3.5** (Part-One of Lemma 5.1 in Bassily et al. (2014) with slight modifications). *Let $n, d \geq 2$ and $\epsilon > 0$. There is a number $n^* = \Omega(\min(n, \frac{d}{\epsilon}))$ such that for any $\epsilon$-differentially private algorithm $\mathcal{A}$, there is a dataset $\mathcal{D} = \{z_1, ..., z_n\} \subset \{\frac{1}{\sqrt{d}}, -\frac{1}{\sqrt{d}}\}^d \cup \{\mathbf{0}\}$ with $\|\sum_{i=1}^{n} z_i\|_2 = n^*$ such that, with probability at least $1/2$ (taken over the algorithm random coins), we have*

$$\|\mathcal{A}(\mathcal{D}) - q(\mathcal{D})\|_2 = \Omega(\min(1, \frac{d}{n\epsilon})), \tag{12}$$

*where $q(\mathcal{D}) = \frac{1}{n} \sum_{i=1}^{n} z_i$.*

Lemma 3.5 basically says that for any $\epsilon$-DP algorithm, it's impossible to for it to estimate the average of some dataset $z_1, ..., z_n$ with accuracy $o(\min(1, \frac{d}{n\epsilon}))$. Using the loss functions defined in Equation (10), Lemma 3.5 and our reduction theorem 3.4, we have the following theorem, whose proof can be found in the appendix.

**Theorem 3.6** (Lower bound for $\epsilon$-differentially private algorithms). *Let $n, d$ be large enough and $\epsilon > 0$. For every $\epsilon$-differentially private algorithm with output $\theta^{priv} \in \mathbb{R}^d$, there is a dataset $\mathcal{D} = \{z_1, ..., z_n\} \subset \{\frac{1}{\sqrt{d}}, -\frac{1}{\sqrt{d}}\}^d \cup \{\mathbf{0}\}$ such that, with probability at least $1/2$ (over the algorithm random coins), we must have that*

$$L(\theta^{priv}; \mathcal{D}) - \min_{\theta \in \mathbb{R}^d} L(\theta; \mathcal{D}) = \Omega(\min(1, \frac{d}{n\epsilon})). \tag{13}$$

## 4 IMPROVED BOUNDS

In this section, we consider lower bounds for approximate DP. We aim to improve previous results in two ways: to tighten the previous lower bounds and extend this bound to any non-euclidean geometry and the unconstrained case. We make the assumption that $2^{-O(n)} < \delta < o(1/n)$. The assumption on $\delta$ is common in the literature, for example, in Steinke & Ullman (2016).

### 4.1 BACKGROUND

We briefly introduce previous lower bounds for constrained DP-ERM and how changing the loss to be $\ell_1$ and domain to be $\ell_\infty$ balls generalizes the lower bounds to the unconstrained non-euclidean case.

As shown in Table 1, Bassily et al. (2014) demonstrates a lower bound $\Omega(\frac{\sqrt{d}}{n\epsilon})$ for the constrained case. They choose $\mathcal{K}$ to be the unit $\ell_2$ ball and the loss function be linear function $\ell(\theta; z) = -\langle \theta, z \rangle$. The empirical loss function is $L(\theta; \mathcal{D}) = -\langle \theta; \sum_{i=1}^{n} z_i/n \rangle$ with minimizer $\theta^* = \frac{\sum_{i=1}^{n} z_i}{\|\sum_{i=1}^{n} z_i\|_2}$. Hence for any solution $\theta$, one has $L(\theta; \mathcal{D}) - L(\theta^*; \mathcal{D}) = \frac{\|\sum_{i=1}^{n} z_i\|_2}{n}(1 - \langle \theta, \theta^* \rangle) \geq \frac{\|\sum_{i=1}^{n} z_i\|_2}{2n}\|\theta - \theta^*\|_2^2$. Therefore one can reduce the lower bound for DP-ERM to the lower bound for estimating the mean of a dataset, for example, the following lemma:

**Lemma 4.1** (Part-Two of Lemma 5.1 in Bassily et al. (2014)). *Let $\epsilon > 0, \delta = o(1/n)$ and $M = \Omega(\min(n, \sqrt{d}/\epsilon))$, for any $(\epsilon, \delta)$-DP algorithm $\mathcal{A}$, there is a dataset $\mathcal{D} = \{z_1, \cdots, z_n\} \subseteq \{-1/\sqrt{d}, 1/\sqrt{d}\}^d$ with $\|\sum_{i=1}^{n} z_i\|_2 \in [M-1, M+1]$ such that with probability at least $1/3$ (taken over $\mathcal{A}$'s random coins), we have*

$$\|\mathcal{A}(\mathcal{D}) - q(\mathcal{D})\|_2 = \Omega(\min(1, \frac{\sqrt{d}}{\epsilon n})),$$

*where $q(\mathcal{D}) = \frac{1}{n} \sum_{i=1}^{n} z_i$.*

In a word, Bassily et al. (2014) uses linear functions as the objective functions and reduces the lower bound on DP-ERM to the lower bound on estimating the mean. Steinke & Ullman (2016) improved the lower bound by a logarithmic factor by using the group privacy technique.

The previous framework fails in the unconstrained and non-euclidean case for two reasons. First, they rely on the $\ell_2$ ball as the domain, which lacks the generalizability to the general $\ell_p$ norm. Second, to generalize the lower bound to the unconstrained case, linear functions are no longer appropriate to be loss functions, as they can take minus infinity values and do not have a global minimum.

To address these concerns, we consider our problem in an $\ell_\infty$ ball and choose the loss function to be $\ell(\theta; z) = \|\theta - z\|_1$. Formally, the loss function used is the following:

$$\ell(\theta; z) = \|\theta - z\|_1, \theta \in \mathbb{R}^d, z \in \{-1, 1\}^d.$$

The convex domain $\mathcal{K}$ is the $\ell_\infty$ unit ball. For any data-set $\mathcal{D} = \{z_1, ..., z_n\}$, we define

$$L(\theta; \mathcal{D}) = \frac{1}{|\mathcal{D}|} \sum_{i=1}^{|\mathcal{D}|} \ell(\theta; z_i) = \frac{1}{|\mathcal{D}|} \sum_{i=1}^{|\mathcal{D}|} \|\theta - z_i\|_1.$$

One main reason for our choice is that $\ell_1$ and $\ell_\infty$ are the "strongest" norms for loss and domain, respectively, implying lower bounds for general $\ell_p$ geometry by the Holder inequality. Moreover, unlike linear functions, the $\ell_1$ loss function can be generalized to the unconstrained case directly. It suffices to figure out how to prove lower bounds for this constrained setting (with $\ell_1$ loss functions in an $\ell_\infty$ unit ball).

Looking into previous lower bounds, such as Bassily et al. (2014) and Steinke & Ullman (2016), one finds that the core idea is two-step: reduce the lower bound of the DP-ERM to the lower bound of mean estimation first, then build the mean estimation by coding theory, particularly the fingerprinting code to be discussed later. In our case, we can not directly reduce the lower bound of the DP-ERM to the lower bound of mean estimation due to the loss function and domain change. In particular, a large mean estimation error does not necessarily imply a large empirical risk.

Consider a simple example. Recall that we want to minimize $L(\theta; \mathcal{D}) = \sum_{i=1}^n \ell(\theta; z_i)/n$ over the $\ell_\infty$ unit ball $\mathcal{K}$, where $\ell(\theta; z) = \|\theta - z\|_1$ and each $z_i \in \{0, 1\}^d$ as the set up before. If $\frac{1}{n} \sum_{i=1}^n z_i = \frac{1}{2}\mathbf{1}$ where $\mathbf{1}$ means the all-one vector, then we know $L(\theta; \mathcal{D})$ is a constant function, equaling to $d/2$ for any $\theta \in \mathcal{K}$. In this example, for a bad estimator $\theta_{\text{bad}}$, even if $\|\theta_{\text{bad}} - \frac{1}{n} \sum_{i=1}^n z_i\|_2$ is large, it can still be a minimizer to the loss function, i.e., $L(\theta_{\text{bad}}; \mathcal{D}) - \min_{\theta \in \mathcal{K}} L(\theta; \mathcal{D}) = 0$.

### 4.2 FINGERPRINTING CODES

Fingerprinting code was first introduced in Boneh & Shaw (1998a), developed and frequently used to demonstrate lower bounds in the DP community Bun et al. (2018); Steinke & Ullman (2016; 2015).

To overcome the challenge discussed before, we slightly modify the definition of the fingerprinting code used in this work.

**Definition 4.2** ($\ell_1$-loss Fingerprinting Code). A $\gamma$-complete, $\gamma$-sound, $\alpha$-robust $\ell_1$-loss fingerprinting code for $n$ users with length $d$ is a pair of random variables $\mathcal{D} \in \{0, 1\}^{n \times d}$ and Trace : $[0, 1]^d \to 2^{[n]}$ such that the following hold:

**Completeness:** For any fixed $\mathcal{M} : \{0, 1\}^{n \times d} \to [0, 1]^d$,

$$\Pr\left[ L(\mathcal{M}(\mathcal{D}); \mathcal{D}) - \min_\theta L(\theta; \mathcal{D}) \leq \alpha d \wedge (\text{Trace}(\mathcal{M}(\mathcal{D})) = \emptyset) \right] \leq \gamma.$$

**Soundness:** For any $i \in [n]$ and fixed $M : \{0, 1\}^{n \times d} \to [0, 1]^d$,

$$\Pr[i \in \text{Trace}(M(\mathcal{D}_{-i}))] \leq \gamma,$$

where $\mathcal{D}_{-i}$ denotes $\mathcal{D}$ with the $i$th row replaced by some fixed element of $\{0, 1\}^d$.

Definition 4.2 is similar to the one in Steinke & Ullman (2016) (See Definition 3.2 in Steinke & Ullman (2016)), except that their requirement of completeness is $\Pr[\|\mathcal{M}(\mathcal{D}) - q(\mathcal{D})\|_1 \leq \alpha d \wedge$

Trace$(\mathcal{M}(\mathcal{D})) = \emptyset] \leq \gamma$. As discussed before, they use the fingerprinting code in their version to build a lower bound on the mean estimation, while we modify the definition and build a lower bound on the DP-ERM under our set-up.

Following the optimal fingerprinting construction Tardos (2008), and subsequent works Bun et al. (2018) Bassily et al. (2014), we have the following result demonstrating the existence of fingerprinting code in our version.

**Lemma 4.3.** *For every $n \geq 1$, and $\gamma \in (0, 1]$, there exists a $\gamma$-complete, $\gamma$-sound, $1/150$-robust $\ell_1$-loss fingerprinting code for $n$ users with length $d$ where*

$$d = O(n^2 \log(1/\gamma)).$$

### 4.3 MAIN RESULT IN EUCLIDEAN GEOMETRY

Similar to Bun et al. (2018), we have the following standard lemma, which allows us to reduce any $\epsilon < 1$ to the $\epsilon = 1$ case without losing generality. The proof is based on the well-known 'secrecy of the sample' lemma from Kasiviswanathan et al. (2011).

**Lemma 4.4.** *For $0 < \epsilon < 1$, a condition $Q$ has sample complexity $n^*$ for algorithms with $(1, o(1/n))$-differential privacy ($n^*$ is the smallest sample size that there exists an $(1, o(1/n))$-differentially private algorithm $\mathcal{A}$ which satisfies $Q$), if and only if it also has sample complexity $\Theta(n^*/\epsilon)$ for algorithms with $(\epsilon, o(1/n))$-differential privacy.*

We apply the group privacy technique in Steinke & Ullman (2016), which needs the following technical lemma:

**Lemma 4.5.** *Let $n, k$ be two large positive integers such that $k < n/1000$. Let $n_k = \lfloor n/k \rfloor$. Let $z_1, \cdots, z_{n_k}$ be $n_k$ numbers where $z_i \in \{0, 1, 1/2\}$ for all $i \in [n_k]$. For any real value $q \in [0, 1]$, if we copy each $z_i$ $k$ times, and append $n - kn_k$ '0' to get $n$ numbers $z'_1, \cdots, z'_n$, then we have*

$$|\sum_{i=1}^{n_k} |q - z_i|/n_k - \sum_{i=1}^{n} |q - z'_i|/n| \leq 3k/n.$$

*Proof.* Without loss of generality, we can assume $z'_{k(i-1)+1} = z'_{k(i-1)+2} = \cdots = z'_{ki} = z_i$, and $z'_{n-kn_k+1} = z'_{kn_k+2} = \cdots = z'_n = 0$. With this observation, we know

$$\begin{aligned}
&|\sum_{i=1}^{n_k} |q - z_i|/n_k - \sum_{i=1}^{n} |q - z'_i|/n| \\
&= |\sum_{i=1}^{n_k} |q - z_i|(1/n_k - k/n) - \sum_{i=n-kn_k+1}^{n} q/n| \\
&\leq |\sum_{i=1}^{n_k} |q - z_i|(1/n_k - k/n)| + |\sum_{i=n-kn_k+1}^{n} q/n| \\
&\leq n_k(\frac{1}{k/n - 1} - \frac{k}{n}) + k/n \leq 3k/n.
\end{aligned}$$

$\square$

This section's main result is the following theorem, which modifies and generalizes the techniques in Steinke & Ullman (2016); Bassily et al. (2014) to reach a tighter bound for the unconstrained case.

**Theorem 4.6** (Lower bound for $(\epsilon, \delta)$-differentially private algorithms). *Let $n, d$ be large enough and $1 \geq \epsilon > 0, 2^{-O(n)} < \delta < o(1/n)$. For every $(\epsilon, \delta)$-differentially private algorithm with output $\theta^{priv} \in \mathbb{R}^d$, there is a data-set $\mathcal{D} = \{z_1, ..., z_n\} \subset \{0, 1\}^d \cup \{\frac{1}{2}\}^d$ such that*

$$\mathbb{E}[L(\theta^{priv}; \mathcal{D}) - L(\theta^\star; \mathcal{D})] = \Omega(\min(1, \frac{\sqrt{d \log(1/\delta)}}{n\epsilon})GC) \tag{14}$$

*where $\ell$ is $G$-Lipschitz w.r.t. $\ell_2$ geometry, $\theta^\star$ is a minimizer of $L(\theta; \mathcal{D})$, and $C = \sqrt{d}$ is the diameter of $\mathcal{K}$ w.r.t. $\ell_2$ geometry, where $\mathcal{K}$ is the unit $\ell_\infty$ ball containing all possible true minimizers and differs from its usual definition in the constrained setting.*

*Remark* 4.7. The dependence on parameters $GC$ makes sense. For example, one can scale the loss function to be $\hat{\ell}(x; z) = \|ax - z\|_1$ for some constant $a \in (0, 1)$, which decreases Lipschitz constant $G$ but increases the diameter $C$ (we should choose $\mathcal{K}$ to contain all possible minimizes).

This bound improves a log factor over Bassily et al. (2014) and can be directly extended to the constrained bounded setting, by setting the constrained domain to be the unit $\ell_\infty$ ball.

### 4.4 Extension to Non-Euclidean Geometry

We illustrate the power of our construction in Theorem 4.6, by showing that the same bound holds for any $\ell_p$ geometry where $p \geq 1$ in the constrained setting, and the bound is tight for all $1 < p \leq 2$, improving/generalizing existing results in Asi et al. (2021); Bassily et al. (2021b). Our construction is advantageous in that it uses $\ell_1$ loss and $\ell_\infty$-ball-like domain in the constrained setting, both being the strongest in their direction when relaxing to $\ell_p$ geometry. Simply using the Holder inequality yields that the product of the Lipschitz constant $G$ and the diameter of the domain $C$ is equal to $d$ when $p$ varies in $[1, \infty)$.

**Theorem 4.8.** *Let $n, d$ be large enough and $1 \geq \epsilon > 0, 2^{-O(n)} < \delta < o(1/n)$ and $p \geq 1$. There exists a convex set $\mathcal{K} \subset \mathbb{R}^d$, such that for every $(\epsilon, \delta)$-differentially private algorithm with output $\theta^{priv} \in \mathcal{K}$, there is a data-set $\mathcal{D} = \{z_1, ..., z_n\} \subset \{0, 1\}^d \cup \{\frac{1}{2}\}^d$ such that*

$$\mathbb{E}[L(\theta^{priv}; \mathcal{D}) - L(\theta^\star; \mathcal{D})] = \Omega(\min(1, \frac{\sqrt{d \log(1/\delta)}}{n\epsilon})GC), \tag{15}$$

*where $\theta^\star$ is a minimizer of $L(\theta; \mathcal{D})$, $\ell$ is $G$-Lipschitz, and $C$ is the diameter of the domain $\mathcal{K}$. Both $G$ and $C$ are defined w.r.t. $\ell_p$ geometry.*

*Proof.* We use the same construction as in Theorem 3.6 which considers $\ell_2$ geometry. We only need to calculate the Lipschitz constant $G$ and the diameter of the domain $\mathcal{K}$.

For the Lipschitz constant $G$, notice that our loss is the $\ell_1$ norm: $\ell(\theta; z) = \|\theta - z\|_1$. It is evident that it is $(d^{1-\frac{1}{p}})$-Lipschitz w.r.t. $\ell_p$ geometry.

For the domain, i.e., the unit $\ell_\infty$ ball $\mathcal{K}$, it obvious that its diameter w.r.t. $\ell_p$ geometry is $C = d^{\frac{1}{p}}$.

To conclude, we find that for any $\ell_p$ geometry where $p \geq 1$, we have $GC = d$ which is independent of $p$. The bound holds for any $\ell_p$ geometry by applying Theorem 4.6. $\qquad\square$

For the unconstrained case, we notice that the optimal $\theta^*$ under our construction must lie in the unit $\ell_\infty$-ball $\mathcal{K} = \{x \in \mathbb{R}^d | 0 \leq x_i \leq 1, \forall i \in [d]\}$, by observing that projecting any point to $\mathcal{K}$ does not increase the $\ell_1$ loss. Therefore, our result can be generalized to the unconstrained case directly. In a word, our result presents lower bounds $\Omega(\frac{\sqrt{d \log(1/\delta)}}{\epsilon n})$ for all $p \geq 1$ and for both constrained case and unconstrained case. Remarkably, our bound is the best for $p$ near 1 and $p > 2$ to our knowledge.

## 5 Conclusion

This paper studies lower bounds for DP-ERM in the unconstrained case and non-euclidean geometry. We propose a simple but powerful black-box reduction approach that can transfer any lower bound in the constrained case to the unconstrained one, indicating that getting rid of dimension dependence is generally impossible. We also prove better lower bounds for approximate-DP ERM for any $\ell_p$ geometry when $p \geq 1$ by considering $\ell_1$ loss over the $\ell_\infty$ ball. Our bound is tight when $1 \leq p \leq 2$ and novel for $p > 2$. However, there is a gap between the current best upper bound Bassily et al. (2021b) and our lower bound when $p > 2$. Filling this gap can be an exciting and interesting problem. Designing better algorithms for general (un)constrained DP-ERM based on our insights would also be an interesting and meaningful direction, which we leave as future work.

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

# A    ADDITIONAL BACKGROUND KNOWLEDGE

## A.1    GENERALIZED LINEAR MODEL (GLM)

The generalized linear model (GLM) is a flexible generalization of ordinary linear regression that allows for response variables with error distribution models other than a normal distribution. To be specific,

**Definition A.1** (Generalized linear model (GLM)).  The generalized linear model (GLM) is a special class of ERM problems where the loss function $\ell(\theta, d)$ takes the following inner-product form:

$$\ell(\theta; z) = \ell(\langle \theta, x \rangle; y) \tag{16}$$

for $z = (x, y)$. Here, $x \in \mathbb{R}^d$ is usually called the feature vector and $y \in \mathbb{R}$ is called the response.

# B    CONSTRUCTION OF FINGERPRINTING CODES

To address the digital watermarking problem, Fingerprinting codes were introduced by Boneh & Shaw (1998b).  Imagine a company selling software to users.  A fingerprinting code is a pair of randomized algorithms (Gen, Trace), where Gen generates a length $d$ code for each user $i$.  To prevent any malicious coalition of users copy and distributing the software, the Trace algorithm can trace one of the malicious users, given a code produced by the coalition of users. They may only can the bits with a divergence in the code: any bit in common is potentially vital to the software and risky to change.

In this section, we introduce the fingerprinting code used by Bun et al. (2018), which is based on the first optimal fingerprinting code Tardos (2008) with additional error robustness. The mechanism of the fingerprinting code is described in Algorithm 1 for completeness.

The sub-procedure part is the original fingerprinting code in Tardos (2008), with a pair of randomized algorithms (Gen, Trace). The code generator Gen outputs a codebook $C \in \{0, 1\}^{n \times d}$. The $ith$ row of $C$ is the codeword of user $i$. The parameter $d$ is called the length of the fingerprinting code.

We make the formal definition of fingerprinting codes:

**Definition B.1** (fingerprinting codes).  Given $n, d \in \mathbb{N}, \xi \in (0, 1]$, a pair of (random) algorithms (Gen, Trace) is called an $(n, d)$-fingerprinting code with security $\xi \in (0, 1]$ if Gen outputs a codebook $C \in \{0, 1\}^{n \times d}$ and for any (possibly randomized) adversary $\mathcal{A}_{FP}$ and any subset $S \subseteq [n]$, if we set $c \leftarrow_R \mathcal{A}_{FP}(C_S)$, then

- $\Pr[c \in F(C_S) \bigwedge \text{Trace}(C, c) = \perp] \leq \xi$

- $\Pr\left[\text{Trace}\,(C, c) \in [n] \backslash S\right] \leq \xi$

where $F\left(C_S\right) = \left\{c \in \{0, 1\}^d \mid \forall j \in [d], \exists i \in S, c_j = c_{ij}\right\}$, and the probability is taken over the coins of Gen, Trace and $\mathcal{A}_{FP}$.

Fingerprint codes imply the hardness of privately estimating the mean of a dataset over $\{0, 1\}^d$. Otherwise, the coalition of users can simply use the rounded mean of their codes to produce the copy. Then the DP-ERM problem can be reduced to privately estimating the mean by using the linear loss whose minimizer is precisely the mean.

The security property of fingerprinting codes asserts that any codeword can be "traced" to a user $i$. Moreover, we require that the fingerprinting code can find one of the malicious users even when they get together and combine their codewords in any way that respects the marking condition. That is, a tracing algorithm Trace takes as inputs the codebook $C$ and the combined codeword $c'$ and outputs one of the malicious users with high probability.

The sub-procedure Gen$'$ first uses a $\sin^2 x$ like distribution to generate a parameter $p_j$ (the mean) for each column $j$ independently, then generates $C$ randomly by setting each element to be 1 with probability $p_j$ according to its location. The sub-procedure Trace$'$ computes a threshold value $Z$ and a 'score function' $S_i(c')$ for each user $i$, then reports $i$ when its score is higher than the threshold.

The main procedure was introduced in Bun et al. (2018), where Gen adds dummy columns to the original fingerprinting code and applies a random permutation.  Trace can first 'undo' the

---

**Algorithm 1** The Fingerprinting Code (Gen, Trace)

---

**Sub-procedure** $\text{Gen}'$:
Let $d = 100n^2 \log(n/\xi)$ be the length of the code.
Let $t = 1/300n$ be a parameter and let $t'$ be such that $sin^2 t' = t$.
**for** $j = 1, ..., d$: **do**
    Choose random $r$ uniformly from $[t', \pi/2 - t']$ and let $p_j = sin^2 r_j$. Note that $p_j \in [t, 1-t]$.
    For each $i = 1, ..., n$, set $C_{ij} = 1$ with probability $p_j$ independently.
**end for**
**Return:** $C$

**Sub-procedure** $\text{Trace}'(C, c')$:
Let $Z = 20n \log(n/\xi)$ be a parameter.
For each $j = 1, ..., d$, let $q_j = \sqrt{(1 - p_j)/p_j}$.
For each $j = 1, ..., d$, and each $i = 1, ..., n$, let $U_{ij} = q_j$ if $C_{ij} = 1$ and $U_{ij} = -1/q_j$ else wise.
**for** each $i = 1, ..., n$: **do**
    Let $S_i(c') = \sum_{j=1}^{d} c'_j U_{ij}$
    Output $i$ if $S_i(c') \geq Z/2$.
    Output $\perp$ if $S_i(c') < Z/2$ for every $i = 1, ..., n$.
**end for**

**Main-procedure** Gen:
Let $C$ be the (random) output of $\text{Gen}'$, $C \in \{0, 1\}^{n \times d}$
Append $2d$ 0-marked columns and $2d$ 1-marked columns to $C$.
Apply a random permutation $\pi$ to the columns of the augmented codebook.
Let the new codebook be $C' \in \{0, 1\}^{n \times 5d}$.
**Return:** $C'$.

**Main-procedure** $\text{Trace}(C, c')$:
Obtain $C'$ from the shared state with Gen.
Obtain $C$ by applying $\pi^{-1}$ to the columns of $C'$ and removing the dummy columns.
Obtain $c$ by applying $\pi^{-1}$ to $c'$ and removing the symbols corresponding to fake columns.
**Return:** $i$ randomly from $\text{Trace}'(C, c)$.

---

permutation and remove the dummy columns, then use $\text{Trace}'$ as a black box. This procedure makes the fingerprinting code more robust in tolerating a small fraction of errors to the marking condition.

In particular, they prove the fingerprinting code Algorithm 1 has the following property.

**Theorem B.2** (Theorem 3.4 in Bun et al. (2018))**.** *For every* $d$, *and* $\gamma \in (0, 1]$, *there exists a* $(n, d)$-*fingerprinting code with security* $\gamma$ *robust to a* $1/75$ *fraction of errors for, for*

$$n = \Omega(\sqrt{d/\log(1/\gamma)})$$

## C   OMITTED PROOF FOR SECTION 3

### C.1   PROOF OF LEMMA 3.5

The proof of Lemma 3.5 is basically the same as the proof in Bassily et al. (2014) with minor modifications. Readers familiar with the literature can feel free to skip it.

*Lemma 3.5.* Let $n, d \geq 2$ and $\epsilon > 0$. There is a number $n^* = \Omega(\min(n, \frac{d}{\epsilon}))$ such that for any $\epsilon$-differentially private algorithm $\mathcal{A}$, there is a dataset $\mathcal{D} = \{z_1, ..., z_n\} \subset \{\frac{1}{\sqrt{d}}, -\frac{1}{\sqrt{d}}\}^d \cup \{\mathbf{0}\}$ with $\|\sum_{i=1}^{n} z_i\|_2 = n^*$ such that, with probability at least $1/2$ (taken over the algorithm random coins), we have

$$\|\mathcal{A}(\mathcal{D}) - q(\mathcal{D})\|_2 = \Omega(\min(1, \frac{d}{n\epsilon})), \quad\quad\quad (17)$$

where $q(\mathcal{D}) = \frac{1}{n} \sum_{i=1}^{n} z_i$.

*Proof.* By using a standard packing argument we can construct $K = 2^{\frac{d}{2}}$ points $z^{(1)}, ..., z^{(K)}$ in $\{\frac{1}{\sqrt{d}}, -\frac{1}{\sqrt{d}}\}^d \cup \{\mathbf{0}\}$ such that for every distinct pair $z^{(i)}, z^{(j)}$ of these points, we have

$$\|z^{(i)} - z^{(j)}\|_2 \geq \frac{1}{8} \tag{18}$$

It is easy to show the existence of such a set of points using the probabilistic method (for example, the Gilbert-Varshamov construction of a linear random binary code).

Fix $\epsilon > 0$ and define $n^\star = \frac{d}{20\epsilon}$. Let's first consider the case where $n \leq n^\star$. We construct $K$ datasets $\mathcal{D}^{(1)}, ..., \mathcal{D}^{(K)}$ where for each $i \in [K]$, $\mathcal{D}^{(i)}$ contains $n$ copies of $z^{(i)}$. Note that $q(\mathcal{D}^{(i)}) = z^{(i)}$, we have that for all $i \neq j$,

$$\|q(\mathcal{D}^{(i)}) - q(\mathcal{D}^{(j)})\|_2 \geq \frac{1}{8} \tag{19}$$

Let $\mathcal{A}$ be any $\epsilon$-differentially private algorithm. Suppose that for every $\mathcal{D}^{(i)}, i \in [K]$, with probability at least $1/2$, $\|\mathcal{A}(\mathcal{D}^{(i)}) - q(\mathcal{D}^{(i)})\|_2 < \frac{1}{16}$, i.e., $Pr[\mathcal{A}(\mathcal{D}^{(i)}) \in B(\mathcal{D}^{(i)})] \geq \frac{1}{2}$ where for any dataset $\mathcal{D}$, $B(\mathcal{D})$ is defined as

$$B(\mathcal{D}) = \{x \in R^d : \|x - q(\mathcal{D})\|_2 < \frac{1}{16}\} \tag{20}$$

Note that for all $i \neq j$, $\mathcal{D}^{(i)}$ and $\mathcal{D}^{(j)}$ differs in all their $n$ entries. Since $\mathcal{A}$ is $\epsilon$-differentially private, for all $i \in [K]$, we have $Pr[A(\mathcal{D}^{(1)}) \in B(\mathcal{D}^{(i)})] \geq \frac{1}{2}e^{-\epsilon n}$. Since all $B(\mathcal{D}^{(i)})$ are mutually disjoint, then

$$\frac{K}{2}e^{-\epsilon n} \leq \sum_{i=1}^{K} Pr[\mathcal{A}(\mathcal{D}^{(1)}) \in B(\mathcal{D}^{(i)})] \leq 1 \tag{21}$$

which implies that $n > n^\star$ for sufficiently large $p$, contradicting the fact that $n \leq n^\star$. Hence, there must exist a dataset $\mathcal{D}^{(i)}$ on which $A$ makes an $\ell_2$-error on estimating $q(\mathcal{D})$ which is at least $1/16$ with probability at least $1/2$. Note also that the $\ell_2$ norm of the sum of the entries of such $\mathcal{D}^{(i)}$ is $n$.

Next, we consider the case where $n > n^\star$. As before, we construct $K = 2^{\frac{p}{2}}$ datasets $\tilde{\mathcal{D}}^{(1)}, \cdots, \tilde{\mathcal{D}}^{(K)}$ of size $n$ where for every $i \in [K]$, the first $n^\star$ elements of each dataset $\tilde{\mathcal{D}}^{(i)}$ are the same as dataset $\mathcal{D}^{(i)}$ from before whereas the remaining $n - n^\star$ elements are $\mathbf{0}$.

Note that any two distinct datasets $\tilde{\mathcal{D}}^{(i)}, \tilde{\mathcal{D}}^{(j)}$ in this collection differ in exactly $n^\star$ entries. Let $\mathcal{A}$ be any $\epsilon$-differentially private algorithm for answering $q$. Suppose that for every $i \in [K]$, with probability at least $1/2$, we have that

$$\|\mathcal{A}(\tilde{\mathcal{D}}^{(i)}) - q(\tilde{\mathcal{D}}^{(i)})\|_2 < \frac{n^\star}{16n} \tag{22}$$

Note that for all $i \in [K]$, we have that $q(\tilde{\mathcal{D}}^{(i)}) = \frac{n^*}{n}q(\mathcal{D}^{(i)})$. Now, we define an algorithm $\tilde{\mathcal{A}}$ for answering $q$ on datasets $\mathcal{D}$ of size $n^\star$ as follows. First, $\tilde{\mathcal{A}}$ appends $\mathbf{0}$ as above to get a dataset $\tilde{\mathcal{D}}$ of size $n$. Then, it runs $\mathcal{A}$ on $\tilde{\mathcal{D}}$ and outputs $\frac{n^*\mathcal{A}(\tilde{\mathcal{D}})}{n}$. Hence, by the post-processing propertty of differential privacy, $\tilde{\mathcal{A}}$ is $\epsilon$-differentially private since $\mathcal{A}$ is $\epsilon$-differentially private. Thus for every $i \in [K]$, with probability at least $1/2$, we have that $\|\tilde{\mathcal{A}}(\mathcal{D}^{(i)}) - q(\mathcal{D}^{(i)})\|_2 < \frac{1}{16}$. However, this contradicts our result in the first part of the proof. Therefore, there must exist a dataset $\tilde{\mathcal{D}}^{(i)}$ in the above collection such that, with a probability at least $1/2$,

$$\|\mathcal{A}(\tilde{\mathcal{D}}^{(i)}) - q(\tilde{\mathcal{D}}^{(i)})\|_2 \geq \frac{n^\star}{16n} \geq \frac{d}{320\epsilon n} \tag{23}$$

Note that the $\ell_2$ norm of the sum of entries of such $\tilde{D}^{(i)}$ is always $n^\star$. $\qquad \square$

## C.2 PROOF OF THEOREM 3.6

The proof does not use the reduction Theorem 3.4 directly as a black box but uses the intuition behind it in detail.

*Theorem 3.6.* Let $n, d \geq 2$ and $\epsilon > 0$. For every $\epsilon$-differentially private algorithm with output $\theta^{priv} \in \mathbb{R}^d$, there is a dataset $\mathcal{D} = \{z_1, ..., z_n\} \subset \{\frac{1}{\sqrt{d}}, -\frac{1}{\sqrt{d}}\}^d \cup \{\mathbf{0}\}$ such that, with probability at least $1/2$ (over the algorithm random coins), we must have that

$$L(\theta^{priv}; \mathcal{D}) - \min_\theta L(\theta; \mathcal{D}) = \Omega(\min(1, \frac{d}{n\epsilon})). \tag{24}$$

*Proof.* We can prove this theorem directly by combining the lower bound in Bassily et al. (2014) and our reduction approach (Theorem 3.4), but we try to give a complete proof as an example to demonstrate how does our black-box reduction approach work out.

Let $\mathcal{A}$ be an $\epsilon$-differentially private algorithm for minimizing $L$ and let $\theta^{priv}$ denote its output, define $r := \theta^{priv} - \theta^*$. First, observe that for any $\theta \in \mathbb{R}^d$ and dataset $\mathcal{D}$ as constructed in Lemma 3.5 (recall that $\mathcal{D}$ consists of $n^*$ copies of a vector $z \in \{\frac{1}{\sqrt{d}}, -\frac{1}{\sqrt{d}}\}^d$ and $n - n^*$ copies of $\mathbf{0}$).

$$L(\theta^*; \mathcal{D}) = \frac{n - n^*}{n} \max\{0, \|\theta^*\|_2 - 1\} + \frac{n^*}{n} \min_{\|y\|_2 \leq 1} (-\langle y, z \rangle + \|\theta^* - y\|_2) = -\frac{n^*}{n} \tag{25}$$

when $\theta^* = z$, and also

$$\begin{aligned}
L(\theta^{priv}; \mathcal{D}) &= \frac{n - n^*}{n} \max\{0, \|\theta^{priv}\|_2 - 1\} + \frac{n^*}{n} \min_{\|y\|_2 \leq 1} (-\langle y, z \rangle + \|\theta^{priv} - y\|_2) \\
&\geq \frac{n^*}{n} \min_{\|y\|_2 \leq 1} (-\langle y, z \rangle + \|\theta^{priv} - y\|_2) \\
&= \frac{n^*}{n} \min_{\|y\|_2 \leq 1} (-\langle y, z \rangle + \|r + z - y\|_2) \\
&\quad \text{(because } \theta^* = z) \\
&\geq \frac{n^* \min\{1, \|r\|_2^2\}}{8n} - \frac{n^*}{n}
\end{aligned}$$

the last inequality follows by discussing the norm of $y - z$. If $\|y - z\|_2 \leq \|r\|_2/2$, then

$$\|r + z - y\|_2 \geq \|r\|_2/2 \geq \min\{1, \|r\|_2^2\}/2 \tag{26}$$

combining with the fact that $|\langle y, z \rangle| \leq 1$ proves the last inequality.

If $\|y - z\|_2 \geq \|r\|_2/2$, then we have $\min_{\|y\|_2 \leq 1} -\langle y, z \rangle \geq -1 + \frac{\|r\|_2^2}{8}$. To prove this, we assume $z = e_1$ without loss of generality and $y - z = (x_1, ..., x_d)$ where $\sum_{i=1}^d x_i^2 \geq \|r\|_2^2/4$. Since $\|y\|_2 = \|y - z + z\|_2 \leq 1$, we must have

$$1 + \sum_{i=1}^d x_i^2 + 2x_1 \leq 1 \tag{27}$$

Thus $-\langle y, z \rangle = -1 - \langle y - z, z \rangle = -1 - x_1 \geq -1 + \|r\|_2^2/8$ as desired, which finishes the discussion on the second case.

From the above result we have that

$$L(\theta^{priv}; \mathcal{D}) - L(\theta^*; \mathcal{D}) \geq \frac{n^* \min\{1, \|r\|_2^2\}}{8n} \tag{28}$$

To proceed, suppose for the sake of a contradiction, that for every dataset $\mathcal{D} = \{z_1, ..., z_n\} \subset \{\frac{1}{\sqrt{d}}, -\frac{1}{\sqrt{d}}\}^d \cup \{\mathbf{0}\}$ with $\|\sum_{i=1}^n z_i\|_2 = n^*$, with probability more than $1/2$, we have that $\|\theta^{priv} - \theta^*\|_2 = \|r\|_2 \neq \Omega(1)$. Let $\tilde{\mathcal{A}}$ be an $\epsilon$-differentially private algorithm that first runs $\mathcal{A}$ on the data and then outputs $\frac{n^*}{n} \theta^{priv}$. Recall that $q(\mathcal{D}) = \frac{n^*}{n} \theta^*$, this implies that for every dataset $\mathcal{D} = \{z_1, ..., z_n\} \subset \{\frac{1}{\sqrt{d}}, -\frac{1}{\sqrt{d}}\}^d \cup \{\mathbf{0}\}$ with $\|\sum_{i=1}^n z_i\|_2 = n^*$, with probability more than $1/2$, $\|\tilde{\mathcal{A}}(\mathcal{D}) - q(\mathcal{D})\|_2 \neq \Omega(\min(1, \frac{d}{n\epsilon}))$ which contradicts Lemma 3.5. Thus, there must exists a dataset $\mathcal{D} = \{z_1, ..., z_n\} \subset \{\frac{1}{\sqrt{d}}, -\frac{1}{\sqrt{d}}\}^d \cup \{\mathbf{0}\}$ with $\|\sum_{i=1}^n z_i\|_2 = n^*$, such that with pr obability more than $1/2$, we have $\|r\|_2 = \|\theta^{priv} - \theta^*\|_2 = \Omega(1)$, and as a result

$$L(\theta^{priv}; \mathcal{D}) - L(\theta^*; \mathcal{D}) = \Omega(\min(1, \frac{d}{n\epsilon})) \tag{29}$$

$\square$

# D    OMITTED PROOF FOR SECTION 4

## D.1    PROOF OF LEMMA 4.3

*Lemma 4.3.* For every $n \geq 1$, and $\gamma \in (0,1]$, there exists a $\gamma$-complete, $\gamma$-sound, $1/150$-robust $\ell_1$-loss fingerprinting code for $n$ users with length $d$ where

$$d = O(n^2 \log(1/\gamma)).$$

*Proof.* We want to find $\alpha$ such that any set satisfying the completeness condition in the above definition is a subset of the $F_\beta$ set of Bun et al. (2018) after rounded to binary numbers, which is

$$F_\beta(\mathcal{D}) = \left\{ c' \in \{0,1\}^d | \Pr_{j \in [d]} [\exists i \in [n], c'_j = \mathcal{D}_{ij}] \geq 1 - \beta \right\}$$

Suppose, round the output $\mathcal{M}(\mathcal{D}) \in [0,1]^d$ to a binary vector $c \in \{0,1\}^d$ where $c \notin F_\beta(\mathcal{D})$, then it makes an "illegal" bit on at least $\beta d$ columns, where each of these columns shares the same number (all-one or all-minus-one columns). It means that on each of these columns, $\mathcal{M}(\mathcal{D})$ has the opposite sign to the shared number, which means on this column, say $i$, the induced loss is lower bounded:

$$\frac{1}{n} \sum_{j=1}^n (|(\mathcal{M}(\mathcal{D})_i - \mathcal{D}_{ij}| - |\text{sign}(\bar{\mathcal{D}}_i) - \mathcal{D}_{ij}|) = \frac{1}{n} \sum_{j=1}^n |(\mathcal{M}(\mathcal{D})_i - \mathcal{D}_{ij}| \geq 1,$$

which means $L(\mathcal{M}(\mathcal{D}); \mathcal{D}) - \min_\theta L(\theta; \mathcal{D}) \geq \beta d/2$. By Theorem B.2 we get $\beta = 1/75$ and conclude our proof. □

## D.2    PROOF OF LEMMA 4.4

*Lemma 4.4.* For $0 < \epsilon < 1$, a condition $Q$ has sample complexity $n^*$ for algorithms with $(1, o(1/n))$-differential privacy ($n^*$ is the smallest sample size that there exists an $(1, o(1/n))$-differentially private algorithm $\mathcal{A}$ which satisfies $Q$), if and only if it also has sample complexity $\Theta(n^*/\epsilon)$ for algorithms with $(\epsilon, o(1/n))$-differential privacy.

*Proof.* The proof uses a black-box reduction, therefore doesn't depend on $Q$. The direction that $O(n^*/\epsilon)$ samples are sufficient is equal to proving the assertion that given a $(1, o(1/n))$-differentially private algorithm $\mathcal{A}$, we can get a new algorithm $\mathcal{A}'$ with $(\epsilon, o(1/n))$-differential privacy at the cost of shrinking the size of the dataset by a factor of $\epsilon$.

Given input $\epsilon$ and a dataset $X$, we construct $A'$ to first generate a new dataset $T$ by selecting each element of $X$ with probability $\epsilon$ independently, then feed $T$ to $\mathcal{A}$. Fix an event $S$ and two neighboring datasets $X_1, X_2$ that differs by a single element $i$. Consider running $\mathcal{A}$ on $X_1$. If $i$ is not included in the sample $T$, then the output is distributed the same as a run on $X_2$. On the other hand, if $i$ is included in the sample $T$, then the behavior of $\mathcal{A}$ on $T$ is only a factor of $e$ off from the behavior of $\mathcal{A}$ on $T \setminus \{i\}$. Again, because of independence, the distribution of $T \setminus \{i\}$ is the same as the distribution of $T$ conditioned on the omission of $i$.

For a set $X$, let $p_X$ denote the distribution of $\mathcal{A}(X)$, we have that for any event $S$,

$$
\begin{aligned}
p_{X_1}(S) &= (1-\epsilon)p_{X_1}(S|i \notin T) + \epsilon p_{X_1}(S|i \in T) \\
&\leq (1-\epsilon)p_{X_2}(S) + \epsilon(e \cdot p_{X_2}(S) + \delta) \\
&\leq \exp(2\epsilon)p_{X_2}(S) + \epsilon\delta
\end{aligned}
$$

A lower bound of $p_{X_1}(S) \geq \exp(-\epsilon)p_{X_2}(S) - \epsilon\delta/e$ can be obtained similarly. To conclude, since $\epsilon\delta = o(1/n)$ as the sample size $n$ decreases by a factor of $\epsilon$, $\mathcal{A}'$ has $(2\epsilon, o(1/n))$-differential privacy. The size of $X$ is roughly $1/\epsilon$ times larger than $T$, combined with the fact that $\mathcal{A}$ has sample complexity $n^*$ and $T$ is fed to $\mathcal{A}$, $\mathcal{A}'$ has sample complexity at least $\Theta(n^*/\epsilon)$.

For the other direction, simply using the composability of differential privacy yields the desired result. In particular, by the $k$-fold adaptive composition theorem in Dwork et al. (2006), we can combine $1/\epsilon$ independent copies of $(\epsilon, \delta)$-differentially private algorithms to get an $(1, \delta/\epsilon)$ one and notice that if $\delta = o(1/n)$, then $\delta/\epsilon = o(1/n)$ as well because the sample size $n$ is scaled by a factor of $\epsilon$ at the same time, offsetting the increase in $\delta$. □

### D.3 PROOF OF THEOREM 4.6

*Theorem 4.6.* Let $n, d$ be large enough and $1 \geq \epsilon > 0, 2^{-O(n)} < \delta < o(1/n)$. For every $(\epsilon, \delta)$-differentially private algorithm with output $\theta^{priv} \in \mathbb{R}^d$, there is a data-set $\mathcal{D} = \{z_1, ..., z_n\} \subset \{0, 1\}^d \cup \{\frac{1}{2}\}^d$ such that

$$\mathbb{E}[L(\theta^{priv}; \mathcal{D}) - L(\theta^\star; \mathcal{D})] = \Omega(\min(1, \frac{\sqrt{d \log(1/\delta)}}{n\epsilon})GC) \tag{30}$$

where $\ell$ is G-Lipschitz w.r.t. $\ell_2$ geometry, $\theta^\star$ is a minimizer of $L(\theta; \mathcal{D})$, and $C = \sqrt{d}$ is the diameter of $\mathcal{K}$ w.r.t. $\ell_2$ geometry, where $\mathcal{K}$ is the unit $\ell_\infty$ ball containing all possible true minimizers and differs from its usual definition in the constrained setting.

*Proof.* Let $k = \Theta(\log(1/\delta))$ be a parameter to be determined later satisfying $k/n < 1/6000$, and $n_k = \lfloor n/k \rfloor$. Consider the case when $d \geq d_{n_k}$ first, where $d_{n_k} = O(\epsilon^2 n_k^2 \log(1/\delta))$.

Without loss of generality, we assume $\epsilon = 1$ due to Lemma 4.4, and $d_{n_k} = O(n_k^2 \log(1/\delta))$ corresponds to the number in Lemma 4.3 where we set $\gamma = \delta$.

We use contradiction to prove that for any $(\epsilon, \delta)$-DP mechanism $\mathcal{M}$, there exists some $\mathcal{D} \in \{0, 1\}^{n \times d}$ such that

$$\mathbb{E}[L(\mathcal{M}(\mathcal{D}); \mathcal{D}) - L(\theta^\star; \mathcal{D})] \geq \Omega(d). \tag{31}$$

Assume for contradiction that $\mathcal{M} : \{0, 1\}^{n \times d} \to [0, 1]^d$ is a (randomized) $(\epsilon, \delta)$-DP mechanism such that

$$\mathbb{E}[L(\mathcal{M}(\mathcal{D}); \mathcal{D}) - L(\theta^\star; \mathcal{D})] < \frac{d}{3000}$$

for all $\mathcal{D} \in \{0, 1\}^{n \times d}$. We then construct a mechanism $\mathcal{M}_k = \{0, 1\}^{n_k \times d}$ with respect to $\mathcal{M}$ as follows: with input $\mathcal{D}^k \in \{0, 1\}^{n_k \times d}$, $\mathcal{M}_k$ will copy $\mathcal{D}^k$ for $k$ times and append enough 0's to get a dataset $\mathcal{D} \in \{0, 1\}^{n \times d}$. The output is $\mathcal{M}_k(\mathcal{D}^k) = \mathcal{M}(\mathcal{D})$. $\mathcal{M}_k$ is $(k, \frac{e^k-1}{e-1}\delta)$-DP by the group privacy.

We consider algorithm $\mathcal{A}_{FP}$ to be the adversarial algorithm in the fingerprinting codes, which rounds the output $\mathcal{M}_k(\mathcal{D}^k)$ to the binary vector, i.e., rounding those coordinates with values no less than 1/2 to 1 and the remaining 0, and let $c = \mathcal{A}_{FP}(\mathcal{M}(\mathcal{D}))$ be the vector after rounding. As $\mathcal{M}_k$ is $(k, \frac{e^k-1}{e-1}\delta)$-DP, $\mathcal{A}_{FP}$ is also $(k, \frac{e^k-1}{e-1}\delta)$-DP.

Considering the $\ell_1$ loss, we can account for the loss caused by each coordinate separately. Recall that $\mathcal{M}_k(\mathcal{D}^k) = \mathcal{M}(\mathcal{D})$. Thus we have that

$$\mathbb{E}[L(\mathcal{M}_k(\mathcal{D}^k); \mathcal{D}^k) - L(\theta^\star; \mathcal{D}^k)]$$
$$= \mathbb{E}[L(\mathcal{M}(\mathcal{D}); \mathcal{D}^k) - L(\theta^\star; \mathcal{D}^k)]$$
$$= \mathbb{E}[L(\mathcal{M}(\mathcal{D}); \mathcal{D}^k)] - \mathbb{E}[L(\mathcal{M}(\mathcal{D}); \mathcal{D})] + L(\theta^\star; \mathcal{D}) - L(\theta^\star; \mathcal{D}^k) + \mathbb{E}[L(\mathcal{M}(\mathcal{D}); \mathcal{D}) - L(\theta^\star; \mathcal{D})]$$
$$\leq 6kd/n + d/3000$$
$$\leq d/900,$$

where we use Lemma 4.5 for the third line.

By Markov Inequality, we know that

$$\Pr[L(\mathcal{M}_k(\mathcal{D}^k); \mathcal{D}^k) - L(\theta^\star; \mathcal{D}^k)] > \frac{d}{150}] \leq 1/5.$$

Lemma 4.3 implies

$$\Pr[L(\mathcal{M}_k(\mathcal{D}^k); \mathcal{D}^k) - L(\theta^\star; \mathcal{D}^k) \leq d/150 \bigwedge \text{Trace}(\mathcal{D}^k, c) = \perp] \leq \delta.$$

By union bound, we can upper bound the probability $\Pr[\text{Trace}(\mathcal{D}^k, c) = \perp] \leq 1/5 + \delta \leq 1/2$. As a result, there exists $i^* \in [n_k]$ such that

$$\Pr[i^* \in \text{Trace}(\mathcal{D}^k, c)] \geq 1/(2n_k). \tag{32}$$

Consider the database with $i^*$ removed, denoted by $\mathcal{D}^k_{-i^*}$. Let $c' = \mathcal{A}_{FP}(\mathcal{M}(\mathcal{D}^k_{-i^*}))$ denote the vector after rounding. By the second property of fingerprinting codes, we have that

$$\Pr[i^* \in \text{Trace}(\mathcal{D}^k_{-i^*}, c')] \le \delta.$$

By the differential privacy and post-processing property of $\mathcal{M}$,

$$\Pr[i^* \in \text{Trace}(\mathcal{D}^k, c)] \le e^k \Pr[i^* \in \text{Trace}(\mathcal{D}^k_{-i^*}, c')] + \frac{e^k - 1}{e - 1}\delta.$$

which implies that

$$\frac{1}{2n_k} \le e^{k+1}\delta. \tag{33}$$

Recall that $2^{-O(n)} < \delta < o(1/n)$, and Equation (33) suggests $k/n \le 2e^k/\delta$ for all valid $k$. But it is easy to see there exists $k = \Theta(\log(1/\delta))$ and $k < n/6000$ to make this inequality false, which is contraction. As a result, there exists some $\mathcal{D} \in \{0, 1\}^{n \times d}$ such that

$$\mathbb{E}[L(\mathcal{M}(\mathcal{D}); \mathcal{D}) - L(\theta^\star; \mathcal{D})] \ge \frac{d}{3000} = \Omega(d).$$

For the $(\epsilon, \delta)$-DP case when $\epsilon < 1$, setting $Q$ to be the condition

$$\mathbb{E}[L(\mathcal{M}(\mathcal{D}); \mathcal{D}) - L(\theta^\star; \mathcal{D})] = O(d)$$

for all $\mathcal{D} \in \{0, 1\}^d$ in Lemma 4.4, we have that any $(\epsilon, \delta)$-DP mechanism $\mathcal{M}$ which satisfies $Q$ for all $\mathcal{D} \in \{0, 1\}^{n \times p}$ must have $n \ge \Omega(\sqrt{d \log(1/\delta)}/\epsilon)$. In another word, for $d \ge O(\epsilon^2 n^2/\log(1/\delta))$, for any $(\epsilon, \delta)$-DP mechanism $\mathcal{M}$, there exists some $\mathcal{D} \in \{0, 1\}^d$ such that

$$\mathbb{E}[L(\mathcal{M}(\mathcal{D}); \mathcal{D}) - L(\theta^\star; \mathcal{D})] \ge \Omega(d).$$

Now we consider the case when $d < d_{n_k}$, i.e., when $n > n^\star \triangleq \Omega(\sqrt{d \log(1/\delta)}/\epsilon)$. Given any dataset $\mathcal{D} \in \{0, 1\}^{n^\star \times d}$, we construct a new dataset $\mathcal{D}'$ based on $\mathcal{D}$ by appending dummy points to $\mathcal{D}$: Specifically, if $n - n^\star$ is even, we append $n - n^\star$ rows among which half are 0 and half are $\{1\}^d$. If $n - n^\star$ is odd, we append $\frac{n-n^\star-1}{2}$ points 0, $\frac{n-n^\star-1}{2}$ points $\{1\}^d$ and one point $\{1/2\}^d$.

Denote the new dataset after appending by $\mathcal{D}'$, we will draw contradiction if there is an $(\epsilon, \delta)$-DP algorithm $\mathcal{M}'$ such that $\mathbb{E}[L(\mathcal{M}(\mathcal{D}'); \mathcal{D}') - L(\theta^\star; \mathcal{D}')] = o(n^\star d/n)$ for all $\mathcal{D}'$, by reducing $\mathcal{M}'$ to an $(\epsilon, \delta)$-DP algorithm $\mathcal{M}$ which satisfies $\mathbb{E}[L(\mathcal{M}(\mathcal{D}); \mathcal{D}) - L(\theta^\star; \mathcal{D})] = o(d)$ for all $\mathcal{D}$.

We construct $\mathcal{M}$ by first constructing $\mathcal{D}'$, and then use $\mathcal{M}'$ as a black box to get $\mathcal{M}(\mathcal{D}) = \mathcal{M}'(\mathcal{D}')$. It's clear that such algorithm for $\mathcal{D}$ preserves $(\epsilon, \delta)$-differential privacy. It suffices to show that if

$$\mathbb{E}[L(\mathcal{M}'(\mathcal{D}'); \mathcal{D}') - L(\theta^\star; \mathcal{D}')] = o(n^\star d/n), \tag{34}$$

then $L(\mathcal{M}(\mathcal{D}); \mathcal{D}) - L(\theta^\star; \mathcal{D}) = o(d)$, which contradicts the previous conclusion for the case $n \le n^\star$. Specifically, if $n - n^\star$ is even, we have that

$$n^\star \mathbb{E}[L(\mathcal{M}(\mathcal{D}); \mathcal{D}) - L(\theta^\star; \mathcal{D})] = n\mathbb{E}[L(\mathcal{M}'(\mathcal{D}'); \mathcal{D}') - L(\theta^\star; \mathcal{D}')].$$

and if $n - n^\star$ is odd, we have that

$$n^\star \mathbb{E}[L(\mathcal{M}(\mathcal{D}); \mathcal{D}) - L(\theta^\star; \mathcal{D})] \le n\mathbb{E}[L(\mathcal{M}'(\mathcal{D}'); \mathcal{D}') - L(\theta^\star; \mathcal{D}')] + d/2,$$

both leading to the desired reduction. We try to explain the above two cases in more detail. If $n - n^*$ is even, then the minimizer of $L(; \mathcal{D})$ and $L(\theta^*; \mathcal{D})$ are the same. And the distributions of the $\mathcal{M}(\mathcal{D})$ and $\mathcal{M}'(\mathcal{D}')$ are identical and indistinguishable. Multiplying $n^*$ or $n$ depends on the number of rows (recall that we normalize the objective function in ERM). The second inequality is because we append one point $\{1/2\}^d$, which can only increase the loss ($\|1/2^d - \theta^*\|_1$) by $d/2$ in the worst case.

Combining results for both cases, we have the following:

$$\mathbb{E}[L(\theta^{priv}; \mathcal{D}) - L(\theta^\star; \mathcal{D})] = \Omega(\min(d, \frac{dn^*}{n})) = \Omega(\min(d, \frac{d\sqrt{d \log(1/\delta)}}{n\epsilon})). \tag{35}$$

Setting Lipschitz constant $G = \sqrt{d}$ and diameter $C = \sqrt{d}$ completes the proof. □

