# OpenReview forum: "Lower Bounds for Differentially Private ERM: Unconstrained and Non-Euclidean"
_ICLR.cc/2023/Conference — Submitted to ICLR 2023_

### Official Review · Reviewer_hdk6 · 2022-10-19

**Confidence:** 4
**Correctness:** 3
**Technical Novelty And Significance:** 2
**Empirical Novelty And Significance:** Not applicable
**Recommendation:** 5

**Clarity, Quality, Novelty And Reproducibility:**

Regarding novelty, the idea of the reduction from constrained to unconstrained optimization is novel in the context of private ERM, although probably classical in optimization. The techniques for the second results are, as already mentioned, pretty standard.

The paper is reasonably clear. In some places the arguments can be written better. For example, equation (7) is not justified. This is easily fixed --- the argument used to prove (9) in fact shows that for every $\theta$, $\tilde{L}(\theta, D) \ge \tilde{L}(\Pi_K \theta, D) = L(\Pi_K \theta, D)$, which implies both (9) and (7).

**Strength And Weaknesses:**

I enjoyed the reduction, which is simple but very neat. The idea is to take a convex 1-Lipschitz loss function $\ell$ defined on a convex domain $K$, and extend it to all of $\mathbb{R}^d$ by taking its inf-convolution with the $\ell_2$ norm. This is a convex 1-Lipschitz loss function it is not hard to show that it induces an ERM problem with the same optimal value and solution.

The second lower bound is less exciting and follows from fingerprinting code techniques that are now standard.

The main issue with the paper is that the paper extends existing lower bounds in relatively minor ways. I am not sure how significant it is to extend ERM lower bounds from constrained to unconstrained optimization. Moreover, it seems like the lower bounds with respect to functions Lipschitz in $\ell_p$ were known for $1 \le p \le 2$, which is also the regime in which these lower bounds are tight. These are both good things to know, but not groundbreaking.

**Summary Of The Paper:**

The paper studies empirical risk minimization with differential privacy and has two main results:

* A simple reduction from constrained to unconstrained minimization that preserves, convexity, the Lipszhitzness constant (with respect to $\ell_2$, and optimal solutions. This reduction allows extending privacy lower bounds from constrained to unconstrained optimization. This implies a lower bound for ERM under pure differential privacy in the unconstrained setting.

* A lower bound of $\Omega\left(\frac{\sqrt{d \log(1/\delta)}}{\varepsilon n}\right)$ for ERM under $(\varepsilon, \delta)$-differential privacy where the loss is the $\ell_1$ norm and optimization is over the cube $[-1, 1]^d$.

**Summary Of The Review:**

The paper contains a neat new reduction, and some good-to-know lower bounds that are of somewhat niche interest. The work is solid but perhaps belongs in a more specialized venue.

---

> ### Author Response · Authors · 2022-11-17
> **Response**
>
> We thank you for your valuable feedback! In practice, many large (high-dimensional) models can achieve good privacy-utility trade-offs, which seems to contradict the constrained case's lower bound.
> Some recent works studying unconstrained cases get dimension-independent results with mild assumptions with supporting experimental evidence.
> This somehow suggests many realistic problems may be more fitting to the unconstrained case.
>
> As for the importance of studying $\ell_p$,
> we already replied to the other two reviewers about it, and we would like to refer you to take a look there.
> We will address your other comments about writing.

---

### Official Review · Reviewer_kgZa · 2022-10-27

**Confidence:** 2
**Correctness:** 3
**Technical Novelty And Significance:** 3
**Empirical Novelty And Significance:** Not applicable
**Recommendation:** 5

**Clarity, Quality, Novelty And Reproducibility:**

In my opinion, the presentation and the organization of the paper should be greatly improved, especially considering that this is a theory paper. I noticed that many terminologies were used without being formally defined. For instance, unconstrained case, non-Euclidean setting, lp geometry, and so on. Frankly, I do not think that these are something well-known to most readers in the machine learning community. Due to the lack of these definitions, I find the paper hard to follow.

**Strength And Weaknesses:**

Strength:

The problem studied in the paper, namely, differentially private empirical risk minimization, is important and has more and more practical applications

Weaknesses:
The authors claimed that one of their main contributions is the establishment of lower bounds under lp norm. However, it is not clear to me why the lp norm is important here. I think this should be clearly clarified when/before summarizing the contributions of the present study. This is also my main concern.



**Summary Of The Paper:**

This paper studied differentially private empirical risk minimization by establishing its lower bounds for the unconstrained case and for the non-euclidean setting. The two main contributions, according to the authors, are as follows: (1) the introduction of a simple but general black-box reduction approach that can transfer any lower bound in the constrained case to the unconstrained case and (2) the establishment of better lower bounds for approximate-DP ERM for any ℓp geometry when p ≥ 1.

**Summary Of The Review:**

see above

---

> ### Author Response · Authors · 2022-11-17
> **Response**
>
> We thank you for your valuable feedback!  The first reviewer also proposed the concern of the importance of studying $\ell_p$.
> We repeat the reply below for convenience and will add more detailed discussions in the new version.
>
> The non-euclidean case has become a recent research hotspot in the field of DP optimization; See, for example, [1,2,3,4,5].
> One important motivation is the wide application of non-euclidean optimization problems, including but not limited to sparse recovery, online learning, combinatorial optimization, and bandits.
> We believe such importance transfers to private ERM and private SCO.
>
> Regarding missing terminologies: sorry for the confusion and we will polish the draft, especially define the terms before we use them as suggested.
>
> [1] Hilal Asi, Vitaly Feldman, Tomer Koren, and Kunal Talwar. Private stochastic convex
> optimization: Optimal rates in L1 geometry. In Marina Meila and Tong Zhang, editors,
> Proceedings of the 38th International Conference on Machine Learning, ICML 2021.
>
> [2]  Raef Bassily, Crist´obal Guzm´an, and Anupama Nandi. Non-euclidean differentially
> private stochastic convex optimization. In Mikhail Belkin and Samory Kpotufe, editors,
> Conference on Learning Theory, COLT 2021.
>
> [3] Raef Bassily, Crist´obal Guzm´an, and Michael Menart. Differentially private stochastic
> optimization: New results in convex and non-convex settings. In Marc’Aurelio Ranzato,
> Alina Beygelzimer, Yann N. Dauphin, Percy Liang, and Jennifer Wortman Vaughan,
> editors, Advances in Neural Information Processing Systems 34.
>
> [4] Yuxuan Han, Zhicong Liang, Zhipeng Liang, Yang Wang, Yuan Yao, and Jiheng Zhang.
> Private streaming sco in ell p geometry with applications in high dimensional online
> decision making. In International Conference on Machine Learning, pages 8249–8279.
> PMLR, 2022.
>
> [5] Gopi S, Lee Y T, Liu D, et al. Private Convex Optimization in General Norms[J]. arXiv preprint arXiv:2207.08347, 2022.

---

### Official Review · Reviewer_9oeX · 2022-10-29

**Confidence:** 4
**Clarity, Quality, Novelty And Reproducibility:** Well written.
**Correctness:** 4
**Technical Novelty And Significance:** 2
**Empirical Novelty And Significance:** Not applicable
**Recommendation:** 5

**Strength And Weaknesses:**

Strengths:
1. The paper is clearly written, easy to understand and the contributions of the paper against the past work has been clearly highlighted.
2. The ideas of extending the lower bounds to the unconstrained case are novel.
3. The paper uses the $\ell_1$ norm to give lower bounds for DP-ERM for a variety of $p$-norms.

Weaknesses:
1. For the jump from constrained to unconstrained case, the assumption of knowing an initial point $\theta_0$ at a distance at most $C$ away from optimal, essentially makes the problem a constrained minimization problem and the rest of the calculations are fairly elementary.
2. For the bounds using $\ell_1$ norm, most of the heavy-lifting is done by previous work.
3. Adding instances of why these geometries are useful and giving algorithms that achieve these bounds can make this paper stronger.
4. It is unclear to me (and I am open to changing this opinion in deliberation with other reviewers and the meta reviewer), how much of a contribution this paper is to the ICLR community. I believe this paper would make an excellent contribution to TMLR since the proofs are correct and it would be nice to have something to cite instead of doing these things yourself as a part of another paper.

**Summary Of The Paper:**

This paper gives lower bounds for differentially private ERM in the unconstrained and non-euclidean case. They provide a simple blackbox reduction approach to reduce lower bounds in the constrained case to unconstrained cases based on the idea of a Lipschitz extension of a function. They also give lower bounds for both the constrained and unconstrained cases by considering $\ell_1$ loss over the $\ell_\infty$ ball.

**Summary Of The Review:**

I think the weaknesses outweigh the strengths and in my opinion the results in this paper are not enough for a full paper contribution. The results are all technically correct, but very incremental and I do not think it is enough to be a full conference paper.

---

> ### Author Response · Authors · 2022-11-17
> **Response**
>
> We thank you for your valuable feedback! Regarding the assumption of knowing $C$: it's a common assumption made by previous works studying DP unconstrained cases, see, for example, [1,2].
> It is even ubiquitous in the (non-private) classic optimization, say it is fair for the algorithm to know a prior about the bound, or it may be difficult to choose a step size.
>
> Regarding the importance of studying $\ell_p$: it has become a recent research hotspot in the field of DP optimization; See, for example, [3,4,5,6,7].
> One important motivation is the wide application of non-euclidean optimization problems, including but not limited to sparse recovery, online learning, combinatorial optimization, and bandits.
> We believe such importance transfers to private ERM and private SCO.
>
> Due to the importance, we think it is good and important for the community to have peer-reviewed lower bounds which are citable.
>
> [1] Shuang Song, Thomas Steinke, Om Thakkar, and Abhradeep Thakurta. Evading the curse of
> dimensionality in unconstrained private glms. In International Conference on Artificial Intelligence
> and Statistics, pp. 2638–2646. PMLR, 2021.
>
> [2] Xuechen Li, Daogao Liu, Tatsunori Hashimoto, Huseyin A Inan, Janardhan Kulkarni,
> Yin Tat Lee, and Abhradeep Guha Thakurta. When does differentially private learning
> not suffer in high dimensions? arXiv:2207.00160, 2022.
>
> [3] Hilal Asi, Vitaly Feldman, Tomer Koren, and Kunal Talwar. Private stochastic convex
> optimization: Optimal rates in L1 geometry. In Marina Meila and Tong Zhang, editors,
> Proceedings of the 38th International Conference on Machine Learning, ICML 2021.
>
> [4]  Raef Bassily, Crist´obal Guzm´an, and Anupama Nandi. Non-euclidean differentially
> private stochastic convex optimization. In Mikhail Belkin and Samory Kpotufe, editors,
> Conference on Learning Theory, COLT 2021.
>
> [5] Raef Bassily, Crist´obal Guzm´an, and Michael Menart. Differentially private stochastic
> optimization: New results in convex and non-convex settings. In Marc’Aurelio Ranzato,
> Alina Beygelzimer, Yann N. Dauphin, Percy Liang, and Jennifer Wortman Vaughan,
> editors, Advances in Neural Information Processing Systems 34.
>
> [6] Yuxuan Han, Zhicong Liang, Zhipeng Liang, Yang Wang, Yuan Yao, and Jiheng Zhang.
> Private streaming sco in ell p geometry with applications in high dimensional online
> decision making. In International Conference on Machine Learning, pages 8249–8279.
> PMLR, 2022.
>
> [7] Gopi S, Lee Y T, Liu D, et al. Private Convex Optimization in General Norms[J]. arXiv preprint arXiv:2207.08347, 2022.

---

### Decision · Program_Chairs · 2023-01-20

**Decision:**

Reject

**Justification For Why Not Higher Score:**

I trust the opinion of Reviewer hdk6 who is a world expert on this topic.

**Justification For Why Not Lower Score:**

The paper has some non-trivial ideas and contributions.

**Metareview: Summary, Strengths And Weaknesses:**

This submission establishes sample complexity lower bounds for differentially private ERM in the unconstrained and non-euclidean cases. A key contribution is a simple blackbox reduction of the constrained case to the unconstrained cases.
The main disadvantage of the submission is that it extends known lower bounds in relatively minor ways. Overall, the reviewers agreed that this work is slightly below the acceptance threshold.